# Evaluation and cost estimation of laboratory test overuse in 43 commonly ordered parameters through a Computerized Clinical Decision Support System (CCDSS) in a large university hospital

**Andrea Tamburrano**[1]*, **Doriana Vallone**[1], **Cinzia Carrozza**[2], **Andrea Urbani**[2], **Maurizio Sanguinetti**[3], **Nicola Nicolotti**[4], **Andrea Cambieri**[4], **Patrizia Laurenti**[1]

**1** Section of Hygiene - Institute of Public Health, Università Cattolica del Sacro Cuore, Roma, Italia, **2** Unit of Biochemical Chemistry and Clinical Molecular Biology, Fondazione Policlinico Universitario A. Gemelli IRCCS, Roma, Italia, **3** Department of Laboratory Sciences and Infectious Diseases, Fondazione Policlinico Universitario A. Gemelli IRCCS, Roma, Italia, **4** Hospital Health Management, Fondazione Policlinico Universitario A. Gemelli IRCCS, Roma, Italia

* dott.tamburrano@gmail.com

## Abstract

### Background

Computerized Clinical Decision Support Systems (CCDSS) have become increasingly important in ensuring patient safety and supporting all phases of clinical decision making. The aim of this study is to evaluate, through a CCDSS, the rate of the laboratory tests over-use and to estimate the cost of the inappropriate requests in a large university hospital.

### Method

In this observational study, hospital physicians submitted the examination requests for the inpatients through a Computerized Physician Order Entry. Violations of the rules in tests requests were intercepted and counted by a CCDSS, over a period of 20 months. Descriptive and inferential statistics (Student's *t*-test and ANOVA) were made. Finally, the monthly comprehensive cost of the laboratory tests was calculated.

### Results

During the observation period a total of 5,716,370 requests were analyzed and 809,245 violations were counted. The global rate of overuse was 14.2% ± 3.0%.

The most inappropriate exams were Alpha Fetoprotein (85.8% ± 30.5%), Chlamydia trachomatis Nucleic Acid Amplification (48.7% ± 8.8%) and Alkaline Phosphatase (20.3% ± 6.5%). The monthly cost of over-utilization was 56,534€ for basic panel, 14,421€ for coagulation, 4,758€ for microbiology, 432€ for immunology exams. All the exams, generated an estimated avoidable cost of 1,719,337€ (85,967€ per month) for the hospital.

**Data Availability Statement:** All relevant data are within the manuscript and its Supporting Information files.

**Funding:** This research did not receive any specific grant from funding agencies in the public, commercial, or not-for-profit sectors.

**Competing interests:** We have no conflicts of interest to report.

## Conclusions

The study confirms the wide variability in over-utilization rates of laboratory tests. For these reasons, the real impact of inappropriateness is difficult to assess, but the generated costs for patients, hospitals and health systems are certainly high and not negligible. It would be desirable for international medical communities to produce a complete panel of prescriptive rules for all the most common laboratory exams that is useful not only to reduce costs, but also to ensure standardization and high-quality care.

## Introduction

The demands of laboratory tests have become the highest volume medical act [1], after years of steady increase. In the United States and Europe, the annual increase in the use of laboratory tests has been around 5% in the last decade. Medicare spending on clinical laboratory tests peaked at almost $ 10 billion, or 1.7% of the total health care budget [2]. Even if the costs of laboratory tests represent less than 5% of hospital spending [3], different studies indicated that pathology investigations are involved in 70–80% of all healthcare decisions [4,5].

According to *Zhi et al.* [6] the inappropriate test can be classified in different forms.

Over-utilization or over-referencing refers to tests ordered but not appropriate, while under-utilization refers to the tests appropriate but not ordered. There are also different types of inadequacy criteria. The objective criteria are clearly defined and independent of the investigation, while the subjective criteria depend on the expert review. Restrictive criteria are required when there is a clear indication for ordering a test, while permissive criteria are required only when there are no contraindications.

A procedure is "appropriate" when it produces more benefits than harm enough to justify its use. Instead, procedures are defined as "equivocal" for which the potential benefits and risks of harm to patients are theoretically equivalent, and "inappropriate" are the procedures for which the risks of harm to the patient clearly outweigh the potential benefits [7].

Computerized Clinical Decision Support Systems (CCDSS) are information technology-based systems that use specific patient characteristics and combine them with a knowledge base using rule-based algorithms [8]. They have become increasingly important in ensuring patient safety and in supporting all phases of clinical decision making. In laboratory medicine, CCDSS are usually used to guide the ordering of tests and diagnostic forecasting by combining informative components and staff skills [9]. By generating reminders or specific patient recommendations that require more appropriate care, CCDSS can also be effective in reducing unnecessary diagnostic tests. In many cases, the assessment is accompanied by an estimate of the savings, often substantial [10–15].

The aim of this study is to evaluate, through a silent CCDSS, the rate of laboratory tests overuse and to estimate the cost of the inappropriate requests in a large university hospital.

## Materials and methods

### Ethics statement

The study is compliant with the Local Ethical Committee Standards of the Fondazione Policlinico Universitario Agostino Gemelli IRCCS; it was approved and registered (Prot. 45189/19 ID: 2849). The study is in accordance with the Helsinki Declaration and EU Regulation 2016/

679 (GDPR) concerning the processing of personal data. For this type of study, Ethical Committee did not foresee the need for participant consent.

## Setting

The *Fondazione Policlinico Universitario Agostino Gemelli IRCCS* is a 1,526-bed high-care-complexity university hospital located in Rome, Italy.

Its laboratory performs about 3.5 million tests every year for inpatients, of about a thousand different types (clinical biochemistry, hematology, coagulation and microbiology). The most frequently requested exams (around 100 types) are performed in the high-automation *Corelab*, a forefront centralized laboratory, and reported on the same day.

During a period of 20 months (from July 2016 to February 2018), we monitored the requests of the most representative laboratory exams made by all the hospital internal departments (except for Emergency, Intensive Care Units and urgent blood test requests). Physicians made inpatient-exam requests through a Computerized Physician Order Entry (CPOE), accessible to medical staff only, that communicated with the Laboratory Information System (LIS) *DNLab* (Dedalus SpA). The laboratory processed the samples, analyzed them and, after validation, automatically sent the results to the clinicians through the LIS.

## Computerized Clinical Decision Support System (CCDSS)

The *Prometeo Appropriatezza* Software (ver. 2.1.3, 2016, *NoemaLife SpA*) intercepted and counted, for each exam, all the laboratory requests and the violations of the rules, in silent mode without blocking or generating pop-ups.

A total of 43 laboratory tests were monitored and 2 different rules have been applied:

- **Biological invariance rules (minimal re-testing intervals)**: each request is verified for the presence (in the same patient) of a still valid result preceding the date of the request acceptance;

- **Incompatibility rules:** each test is associated with a list of incompatible laboratory tests and a list of related exams that must be requested simultaneously.

39 tests for biological invariance, 4 tests for incompatibility rules were monitored.

The rules, principally based on minimal re-testing intervals criteria reported in the most recent international guidelines on the prescription appropriateness for laboratory tests [16,17], were shared and approved, over a period of 1 month, by an expert panel made up of laboratory physicians, hospital department chiefs and hospital management.

Laboratory tests were also grouped into 4 different categories: basic panel, coagulation, immunology, microbiology.

Among basic panel exams, electrolytes, lipid and liver panel, among microbiology exams, culture and antibodies tests were considered as sub-categories, respectively.

All the examined tests have been ordered and counted as single tests, except for Complete Blood Count that included: Red Blood Cells (RBC), Hemoglobin (Hb), Hematocrit (Ht), Mean Corpuscular Volume (MCV), Mean Corpuscular Hemoglobin (MCH), Mean Corpuscular Hemoglobin Concentration (MCHC), Red Cells Dispersion Width (RDW), Platelets (PLTS). This panel of tests was considered and counted in bulk.

Table 1 shows the list of the monitored laboratory tests, their division into categories and sub-categories, and the rules applied.

**Table 1. Laboratory tests, categories, sub categories and rules.**

| Laboratory test | Category | Sub-category | Rule | Rule description |
|---|---|---|---|---|
| Amylase | Basic panel | - | Incompatibility | Test cannot be requested with lipase test |
| Sedimentation Rate | Basic panel | - | Biological invariance | Test was already requested in 7 days |
| White blood cell count and differential | Basic panel | - | Biological invariance | Test was already requested in 7 days |
| Complete Blood Count* | Basic panel | - | Biological invariance | Test was already requested in 7 days |
| Triglycerides | Basic panel | Lipid panel | Biological invariance | Test was already requested in 21 days |
| Protein Electrophoresis | Basic panel | - | Biological invariance | Test was already requested in 21 days |
| Total Cholesterol | Basic panel | Lipid panel | Biological invariance | Test was already requested in 21 days |
| Magnesium | Basic panel | Electrolytes | Biological invariance | Test was already requested in 24 hours |
| Sodium | Basic panel | Electrolytes | Biological invariance | Test was already requested in 24 hours |
| Potassium | Basic panel | Electrolytes | Biological invariance | Test was already requested in 24 hours |
| Chloride | Basic panel | Electrolytes | Biological invariance | Test was already requested in 24 hours |
| Creatinine | Basic panel | - | Biological invariance | Test was already requested in 24 hours |
| Aspartate Aminotransferase | Basic panel | Liver panel | Biological invariance | Test was already requested in 24 hours |
| Alanine Transaminase | Basic panel | Liver panel | Biological invariance | Test was already requested in 24 hours |
| Alkaline Phosphatase | Basic panel | - | Biological invariance | Test was already requested in 24 hours |
| Bilirubin | Basic panel | Liver panel | Biological invariance | Test was already requested in 24 hours |
| Albumin | Basic panel | - | Biological invariance | Test was already requested in 24 hours |
| Total Serum Protein | Basic panel | - | Biological invariance | Test was already requested in 24 hours |
| Gamma-Glutamyl Transferase | Basic panel | Liver panel | Biological invariance | Test was already requested in 24 hours |
| Lactate Dehydrogenase | Basic panel | - | Biological invariance | Test was already requested in 24 hours |
| HDL Cholesterol | Basic panel | Lipid panel | Biological invariance | Test was already requested in 21 days |
| LDL Cholesterol | Basic panel | Lipid panel | Biological invariance | Test was already requested in 21 days |
| HIV1-2 Antibodies | Microbiology | Antibody | Biological invariance | Previous test with positive result |
| Toxoplasma Antibody (IgG) | Microbiology | Antibody | Biological invariance | Previous test with positive result |
| Cytomegalovirus Antiboy (IgG) | Microbiology | Antibody | Biological invariance | Previous test with positive result |
| Epstein-Barr Virus Antibody (IgG) | Microbiology | Antibody | Biological invariance | Previous test with positive result |
| Rubella Antibody (IgG) | Microbiology | Antibody | Biological invariance | Previous test with positive result |

(*Continued*)

**Table 1.** (Continued)

| Laboratory test | Category | Sub-category | Rule | Rule description |
|---|---|---|---|---|
| Chlamydia trachomatis nucleic acid amplification (NAATs) | Microbiology | Culture | Incompatibility | Test cannot be requested on vaginal swab/secretion |
| Blood Culture—Aerobic | Microbiology | Culture | Biological invariance | Test was already requested in 24 hours |
| Blood Culture—Anaerobic | Microbiology | Culture | Biological invariance | Test was already requested in 24 hours |
| Antinuclear Antibody | Immunology | - | Biological invariance | Test was already requested in 90 days |
| Immunoglobulin A | Immunology | - | Biological invariance | Test was already requested in 21 days |
| Immunoglobulin G | Immunology | - | Biological invariance | Test was already requested in 21 days |
| Immunoglobulin M | Immunology | - | Biological invariance | Test was already requested in 21 days |
| Prothrombin time | Coagulation | - | Biological invariance | Test was already requested in 24 hours with results in the normal range |
| Partial Thromboplastin Time | Coagulation | - | Biological invariance | Test was already requested in 24 hours with results in the normal range |
| Fibrinogen | Coagulation | - | Biological invariance | Test was already requested in 24 hours with results in the normal range |
| D-Dimer | Coagulation | - | Biological invariance | Test was already requested in 24 hours with results in the normal range |
| Antithrombin III | Coagulation | - | Biological invariance | Test was already requested in 24 hours with results in the normal range |
| Beta HCG | Other | - | Incompatibility | Test cannot be requested for male subjects** |
| Alpha Fetoprotein (AFP) | Other | - | Incompatibility | Test cannot be requested with nonspecific tumor markers (CEA, CA125, CA19-9, CA15-3, TPA) |
| Procalcitonin | Other | - | Biological invariance | Test was already requested in 24 hours |
| Vitamin B12 | Other | - | Biological invariance | Test was already requested in 30 days |

* Complete Blood Count includes: Red Blood Cells (RBC), Hemoglobin (Hb), Hematocrit (Ht), Mean Corpuscular Volume (MCV), Mean Corpuscular Hemoglobin (MCH), Mean Corpuscular Hemoglobin Concentration (MCHC), Red Cells Dispersion Width (RDW), Platelets (PLTS).

** The test can be requested in male patients only for the diagnosis and monitoring of testicular seminoma.

## Analysis

Descriptive statistics were performed analyzing, over a period of 20 months for each laboratory exam, the monthly mean and standard deviation of requests and violations of the rules. Based on them, the overuse rates were calculated.

Moreover, Student's *t*-test and ANOVA were used to assess differences between quantitative variables. The level of significance was set at 0.05. Statistical analyses were conducted with STATA software ver. 13.1 (Statacorp, College Station, TX, USA).

According to the Italian National Health System (NHS), all treatments were carried out free of charge for the hospitalized patients. Hospitalization costs (including laboratory tests) were reimbursed by the NHS according to Medicare Diagnosis Related Groups (MS-DRGs) [18]. The monthly comprehensive cost of the laboratory tests was calculated in euro (€), according to the 2019 reimbursement fees of the Lazio Region.

## Results

### Overuse

During the observation period, a total of 5,716,370 requests were analyzed (285,819 per month) and 809,245 violations were counted (40,462 per month). The global rate of overuse was 14.2% ± 3.0%.

The rate was 15.2% ± 3.3% for basic panel, 8.1% ± 3.4% for microbiology, 7.2% ± 1.9% for immunology, 5.8% ± 1.1% for coagulation laboratory tests. The overall difference among groups was significant (p<0.001, ANOVA).

Among basic panel exams, the rate was 18.8% ± 4.9% for liver panel, 9.0% ± 2.3% for lipid panel, 16.0% ± 3.8% for electrolytes. The overall difference among sub-groups was significant (p<0.001, ANOVA).

Among microbiology exams, the rate was 8.4% ± 1.7% for cultural tests, 7.7% ± 12.9% for antibodies. No significant differences were observed among sub-groups (p = 0.811, *t*-test).

The most inappropriate exams were Alpha Fetoprotein (85.8% ± 30.5%), Chlamydia trachomatis Nucleic Acid Amplification (48.7% ± 8.8%) and Alkaline Phosphatase (20.3% ± 6.5%). The most appropriate exams were Sedimentation Rate (0.8% ± 0.5%), HIV1-2 Antibodies (2.7% ± 1.6%) and Total Cholesterol (3.4% ± 1.3%).

Table 2 shows the number of monthly requests and the rate of overuse for each monitored exam.

### Cost evaluation

All the exams, globally considered, generated an estimated avoidable cost of 1,719,337€ (85,967€ per month) for the hospital.

The monthly cost of over-utilization was 56,534€ for basic panel, 14,421€ for coagulation, 4,758€ for microbiology, 432€ for immunology exams.

The greatest monthly cost of overuse was for Complete Blood Count (11,334€), Fibrinogen (10,238€) and Total Serum Protein (8,522€) that cover 35% of the total over-utilization cost. The least monthly cost of overuse was for Sedimentation Rate (24€), Antinuclear Antibody (73€) and Immunoglobulin A (99€).

Table 2 shows the unit cost and the total monthly cost for each monitored exam.

## Discussion

This study evaluated the overuse of laboratory tests providing data on the prescriptive activity of a large university hospital (5,716,370 requests) over a long period of time (20 months).

Several authors show variable rates of inappropriateness in laboratory tests: 4.5–95.0% [19], 25.0–75.0% [20] 5.0–95.0% [21], 45.4–93.9% [22]. *Zhi et al.* [6] reported that this variability is due to the great variety in tests, clinical settings, timing (initial vs. repeat testing), adopted criteria (restrictive vs. permissive, subjective vs. objective) and test volume (low-volume vs. high-volume).

Unlike the most recent works on laboratory overuse (compared to which it is necessary to consider a reasonable variability in terms of settings and tests considered), our rate (14.2%) was lower than reported by other authors: *Zhi et al.* (20.6%) [6], *Feldhammer et al.* (27.0%) [23].

The basic panel exams showed an over-utilization rate (15.2%) similar to that reported by *Zhi et al.* (10,2–19,1%) [6] and *May et al.* (11,5%) [24]. Differently, *Rao et al.* reported an higher rate (38%) [25]. Electrolytes registered a rate (16.0%) lower than reported by *Wang et al.* (31.0–40.0%) [26].

**Table 2. Mean and standard deviation of monthly requests and over-utilization rate, unit cost and monthly avoidable cost, for each monitored exam.**

| Exam | Monthly requests | Over-utilization rate (%) | Unit cost (€) | Total monthly cost (€) |
|---|---|---|---|---|
| Alanine Transaminase | 17,361 ± 2,144 | 18.0 ± 4.0 | 1.00 | 3,121 |
| Albumin | 11,302 ± 1,403 | 19.3 ± 4.8 | 1.42 | 3,086 |
| Alkaline Phosphatase | 8,978 ± 1,092 | 20.3 ± 6.5 | 1.04 | 1,898 |
| Alpha Fetoprotein | 176 ± 31 | 85.8 ± 30.5 | 7.40 | 1,103 |
| Amylase | 7,394 ± 800 | 19.4 ± 1.6 | 1.84 | 2,640 |
| Antinuclear Antibody | 115 ± 23 | 6.6 ± 4.2 | 9.56 | 73 |
| Antithrombin III | 1,382 ± 205 | 9.3 ± 2.1 | 5.02 | 640 |
| Aspartate Aminotransferase | 4,482 ± 902 | 19,5 ± 8.0 | 1.04 | 905 |
| Beta HCG | 405 ± 48 | 5.9 ± 1.3 | 6.02 | 144 |
| Bilirubin | 16,239 ± 1,921 | 19.1 ± 4.5 | 1.41 | 4,371 |
| Blood Culture—Aerobic | 1,692 ± 226 | 3.4 ± 1.1 | 13.86 | 809 |
| Blood Culture—Anaerobic | 1,692 ± 226 | 12.1 ± 2.9 | 13.86 | 2,841 |
| Chlamydia trachomatis nucleic acid amplification (NAATs) | 56 ± 39 | 48.7 ± 8.8 | 9.41 | 247 |
| Chloride | 6,316 ± 934 | 19.8 ± 6.5 | 1.13 | 1,395 |
| Complete Blood Count* | 23,604 ± 2,873 | 15.3 ± 3.7 | 3.17 | 11,334 |
| Creatinine | 21,159 ± 2,553 | 17.1 ± 2.9 | 1.13 | 4,084 |
| Cytomegalovirus Antibody (IgG) | 275 ± 70 | 10.4 ± 17.7 | 8.07 | 234 |
| D-Dimer | 1,592 ± 208 | 12.8 ± 2.9 | 4.99 | 1,013 |
| Epstein-Barr Virus Antibody (IgG) | 172 ± 49 | 9.6 ± 18.0 | 12.45 | 202 |
| Fibrinogen | 10,862 ± 1,276 | 7.8 ± 1.5 | 12.18 | 10,238 |
| Gamma-Glutamyl Transferase (GGT) | 10,016 ± 1,431 | 19.1 ± 6.2 | 1.13 | 2,149 |
| HDL Cholesterol | 2,089 ± 487 | 8.5 ± 1.9 | 1.43 | 252 |
| HIV1-2 Antibodies | 350 ± 93 | 2.7 ± 1.6 | 10.90 | 106 |
| Immunoglobulin A | 351 ± 82 | 5.8 ± 2.2 | 4.99 | 99 |
| Immunoglobulin G | 338 ± 78 | 8.1 ± 2.1 | 4.99 | 134 |
| Immunoglobulin M | 327 ± 77 | 7.8 ± 1.9 | 4.99 | 125 |
| Lactate Dehydrogenase | 12,710 ± 1,333 | 19.1 ± 4.8 | 1.13 | 2,748 |
| LDL Cholesterol | 1,857 ± 454 | 9.8 ± 2.0 | 0.67 | 120 |
| Magnesium | 6,865 ± 818 | 12.0 ± 3.7 | 1.55 | 1,301 |
| Partial Thromboplastin Time | 10,961 ± 1,265 | 3.7 ± 3.0 | 2.85 | 1,122 |
| Potassium | 19,619 ± 2,217 | 18.0 ± 4.0 | 1.02 | 3,608 |
| Procalcitonin | 2,051 ± 257 | 12.1 ± 3.0 | 14.41 | 3,611 |
| Protein Electrophoresis | 1,896 ± 315 | 8.3 ± 1.6 | 4.23 | 667 |
| Prothrombin Time | 11,200 ± 1,276 | 4.4 ± 3.0 | 2.85 | 1,408 |
| Rubella Antibody (IgG) | 308 ± 102 | 9.8 ± 19.2 | 7.88 | 215 |
| Sedimentation Rate | 1,596 ± 377 | 0.8 ± 0.5 | 1.95 | 24 |
| Sodium | 19,548 ± 2,211 | 14.2 ± 3.4 | 1.02 | 2,841 |
| Total Cholesterol | 8,081 ± 1,390 | 3,4 ± 1.3 | 1.04 | 273 |
| Total Serum Protein | 10,794 ± 1,447 | 18.8 ± 4.9 | 4.23 | 8,522 |
| Toxoplasma Antibody (IgG) | 163 ± 41 | 8.1 ± 18.8 | 7.79 | 104 |
| Triglycerides | 6,833 ± 1,251 | 15.5 ± 4.0 | 1.17 | 1,219 |
| Vitamin B12 | 638 ± 135 | 6.9 ± 1.8 | 7.32 | 318 |
| White blood cell count and differential | 21,974 ± 2,539 | 5.5 ± 1.7 | 3.91 | 4,619 |
| Total | 285,819 | 14.2 ± 3.0 | - | 85,967 |

* Complete Blood Count includes: Red Blood Cells (RBC), Hemoglobin (Hb), Hematocrit (Ht), Mean Corpuscular Volume (MCV), Mean Corpuscular Hemoglobin (MCH), Mean Corpuscular Hemoglobin Concentration (MCHC), Red Cells Dispersion Width (RDW), Platelets (PLTS).

Microbiology tests registered a rate (8.1%) lower than reported by and *Zhi et al*. (23.1%) [6]. Antibodies showed a rate (7.7%) lower than reported by *Crump et al*. (25.0%) [27].

Coagulation tests registered a rate (5.8%) lower than reported by *Iturratte et al*. (19.7%) [10].

As reported by other authors, we also registered an evident variability between single exams (85% of Alpha Fetoprotein to 0.8% of Sedimentation Rate) and groups (15.2% of basic panel to 5.8% of coagulation tests).

The plurality of medical specialties and hospital policies, as well as health, legislative, economic and political context could explain the variability in inappropriateness found in different countries.

Regarding costs, for the only 43 exams considered (to which only one rule has been applied), we estimated consistent economic savings for the hospital (85,967€ per month). The highest cost was due to basic panel exams, in particular Complete Blood Count, Fibrinogen and Total Serum Proteins, that covered 35% of the total costs for laboratory over-utilization. Antibodies (in microbiology and immunology categories) and Immunoglobulins (in immunology category) had the least impact on the total costs.

In addition to the costs, inappropriateness may also affect other aspects of the health care, such as additional procedures or treatments based on redundant tests, avoidable medical errors, waste of time for doctors, work overload for the laboratory, delays in reporting times [28,29].

In a context of high incidence of medical disputes [30,31], the use of defensive medicine has generated an increasing number of tests performed per patient, usually generated through predetermined panels of tests, that are easily ordered [32]. Utilization and decisions based on routine tests, given the widespread nature of their use, the ease of measurement and low cost, are not always appropriate, leading to undesirable consequences for the patients. An inappropriate use of routine tests minimizes their utility and favors erroneous interpretations, increasing safety risks for the patient. This practice could also lead to order potentially harmful complementary tests, to rule out or corroborate the results obtained.

Additionally, uncertain results and test repetition could cause anxiety in the patients, as they are faced with the possibility of an uncertain diagnosis [33]. For these reasons, the use of routine tests should be based on clear and sufficient scientific evidence.

According to *Lanzoni et al*. [15], although the definition of test panels for the diagnosis/ monitoring of different health conditions is a good tool to increase the clinical governance, a deep evaluation must be done on the real needs on requiring the same tests all the time for a single patient.

Safety risks of inappropriateness should be analysed to implement appropriate strategies to improve their correct use.

## Strengths and limitations

Only a small part of the total performed test types (over 1000) were monitored, as well as the potential rules applicable to each exam are much more varied and numerous than those reported in this study. For this reason, the rate of overuse and the sum of avoidable costs were underestimated. However, the most representative exams and rules (applicable in the context of an automatic CCDSS) were included and fully monitored with over 5 million of laboratory tests performed over a period of 20 months.

The CCDSS counted violations of the over-utilization rules without blocking or generating pop-ups and physicians were not alerted. Although it was not possible to evaluate CCDSS effectiveness in reducing the over-utilization rate, we were able to assess the rate of laboratory

tests overuse without the confounding factor of physician's awareness of a control system on laboratory requests. For the cost estimation we assumed the ideal 100% reduction in tests overuse.

## Conclusions

A recent systematic review compared strategies to change the behavior of doctors and to reduce the inappropriateness rate of the laboratory tests [34]. Various interventions, such as educational strategies, feedback, audit, modification of test modules and reminders (a form of CCDSS) should be planned because effective [35].

This study confirms the wide variability in over-utilization of laboratory tests. For these reasons, the real impact of inappropriateness is difficult to assess, but the generated costs for patients, hospitals and health systems are certainly high and not negligible [28,36,37].

It would be desirable for international medical communities to produce a complete panel of prescriptive rules for all the most common laboratory exams (which consider the variability of settings and clinical conditions) that is useful not only to reduce costs, but also to ensure standardization and high-quality care.

## Supporting information

**S1 Dataset.**
(XLSX)

## Author Contributions

**Conceptualization:** Andrea Tamburrano, Cinzia Carrozza, Andrea Urbani, Patrizia Laurenti.

**Data curation:** Andrea Tamburrano, Cinzia Carrozza, Nicola Nicolotti.

**Formal analysis:** Andrea Tamburrano, Nicola Nicolotti.

**Funding acquisition:** Patrizia Laurenti.

**Investigation:** Andrea Tamburrano, Cinzia Carrozza, Nicola Nicolotti.

**Methodology:** Andrea Tamburrano, Doriana Vallone, Cinzia Carrozza, Andrea Urbani, Nicola Nicolotti, Andrea Cambieri, Patrizia Laurenti.

**Project administration:** Andrea Cambieri, Patrizia Laurenti.

**Resources:** Patrizia Laurenti.

**Software:** Cinzia Carrozza, Nicola Nicolotti.

**Supervision:** Andrea Tamburrano, Andrea Urbani, Maurizio Sanguinetti, Nicola Nicolotti, Andrea Cambieri, Patrizia Laurenti.

**Validation:** Doriana Vallone, Cinzia Carrozza.

**Visualization:** Doriana Vallone, Nicola Nicolotti.

**Writing – original draft:** Andrea Tamburrano, Doriana Vallone.

**Writing – review & editing:** Andrea Tamburrano, Cinzia Carrozza.

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
