## [Decision Letter · Decision Letter 0]

7 Feb 2020

PONE-D-20-01565

Evaluation and cost estimation of inappropriateness in laboratory tests through a Computerized Clinical Decision Support System (CCDSS) in a large university hospital.

PLOS ONE

Dear Dr. Tamburrano,

Thank you for submitting your manuscript to PLOS ONE. After careful consideration, we feel that it has merit but does not fully meet PLOS ONE’s publication criteria as it currently stands. Therefore, we invite you to submit a revised version of the manuscript that addresses the points raised during the review process.

In addition to the issues raised by the reviewers, please address the following.

The use (misuse) of laboratory tests is a common issue that not only affect the laboratories but also create unnecessary work for clinicians that may harm the patients.

As far as I can understand, is the title of the paper wrong, the authors focus only on repeating test, not the appropriateness? For example is an ASAT test seldom relevant.

Furthermore is only a fraction of test investigated. I miss the most frequently requested test, hemoglobin. These factors may explain the relative low frequency of misuse, I agreement with the literature is my observation that the correct number is around 60-70%.

We would appreciate receiving your revised manuscript by Mar 23 2020 11:59PM. To enhance the reproducibility of your results, we recommend that if applicable you deposit your laboratory protocols in protocols.io, where a protocol can be assigned its own identifier (DOI) such that it can be cited independently in the future. For instructions see: http://journals.plos.org/plosone/s/submission-guidelines#loc-laboratory-protocols

We look forward to receiving your revised manuscript.

Kind regards,

Pal Bela Szecsi, M.D. D.M.Sci.

Academic Editor

PLOS ONE

Journal Requirements:

Reviewers' comments:

Reviewer's Responses to Questions

**Comments to the Author**

1. Is the manuscript technically sound, and do the data support the conclusions?

Reviewer #1: Partly

Reviewer #2: Partly

2. Has the statistical analysis been performed appropriately and rigorously? 

Reviewer #1: Yes

Reviewer #2: N/A

3. Have the authors made all data underlying the findings in their manuscript fully available?

Reviewer #1: No

Reviewer #2: Yes

4. Is the manuscript presented in an intelligible fashion and written in standard English?

Reviewer #1: Yes

Reviewer #2: Yes

5. Review Comments to the Author

Reviewer #1: The authors have performed an observational study on the appropriateness of laboratory test orders in a large university hospital. Inappropriate testing is an important issue in health care systems facing increasing demands and diminishing resources.

There might be more information in the data set to be presented. I also have some concerns, mainly regarding the description of the setting.

Major comments:

1) The economic setting should be thoroughly described and discussed because economic incentives have an impact on test ordering behavior. A description of who is paying for the analyses is essential to understand the setting. Is it the patient paying anything (a fixed fee or the total cost)? Does the ordering unit pay to the laboratory, or is the laboratory completely budget financed? Are the hospital and the laboratory owned by the state, a company, or a trust?

2) Does the laboratory accept requests from primary health care centers? Were samples from non-hospital units included?

3) The cost in this paper is defined as the reimbursement fee set by the authorities. The reimbursement system should be explained, allowing the reader to understand the context. The reimbursement fee seems to includes reagents, staff, rental costs of the laboratory site, overhead costs, and so on. Most of the costs, excluding the reagent costs, are more or less constant so that the net savings might be substantially lower than the figures described in the paper. Who would save money with more appropriate testing? Would it have a negative impact on the laboratory economy?

4) Which staff categories are allowed to order laboratory tests? Is it only physicians, or are nurses also allowed to order tests?

5) Were there any temporary physicians from staffing companies? Physicians from staffing companies tend to order more tests.

6) It has previously been described that the demographics of the physicians are of importance for test ordering behavior. Were there any differences between violation rates between medical interns, residents, and consultants? The large number of requests would probably allow for subgroup analysis, which would add substantial value to the paper.

7) This study reports a lower proportion of inappropriate tests than previous studies. Was there any information about the study available to the hospital staff? Was the staff aware of the rules of appropriate testing? Was there any education about this before or during the study? If so, this should be described and discussed, as information and education have been shown to impact test ordering patterns.

8) The use of test panels in the study setting should be described in more detail. Are the tests usually ordered in bundled panels, or as single tests?

9) How were tests included in panels counted towards the total test count? Was the panel counted as one test, or are all the included analytes counted as separate tests?

Minor comments:

1) The Conclusion section could be shortened, and some of the text in this section would be more suitable in the Discussion section.

Reviewer #2: The authors have collected a remarkable number of datapoints regarding inappropriately ordered laboratory tests. They impressively show the financial benefits of addressing this issue.

I have some major comments which need to be addressed imho in order to improve the ms.

The term „Inappropriateness” is used icorrectly. In order to cover all inappropriate lab orders, those underused have to be included. To do so, mostly the anamnesis, physical examination and indication for testing is needed (i.e. signs of mucosal bleeding and positive family history but only APTT is ordered). As this is not presented in this study, I suggest rephrasing. Additionally, overuse does not only consist of tests ordered too soon after baseline measurement (re-testing interval). There are several other causes for laboratory overuse.

See:

Greenberg J, Green JB. Over-testing: why more is not better. Am J Med 2014; 127:362-363.

Cadamuro J, Gaksch M, Wiedemann H, et al. Are laboratory tests always needed? Frequency and causes of laboratory overuse in a hospital setting. Clin Biochem 2018; 54:85-91.

Fryer AA, Smellie WSA. Managing demand for laboratory tests: a laboratory toolkit. J Clin Pathol 2013; 66:62-72.

Cadamuro J, Ibarz M, Cornes M, et al. Managing inappropriate utilization of laboratory resources. Diagnosis (Berl) 2019; 6:5-13.

van Walraven C, Naylor CD. Do we know what inappropriate laboratory utilization is? A systematic review of laboratory clinical audits. Jama 1998; 280:550-558.

Throughout the entire manuscript is reads as if the authors have identified ALL inappropriate testing, which is by not true. This is also the reason why percentages differ compared to those of other authors.

The title reads as if all parameters from the labs portfolio were considered in this study, which is also not true. Therefore, I would also suggest changing the title to reflect the content more precisely: Evaluation and cost estimation of inappropriateness in selected laboratory tests through a Computerized Clinical Decision Support System (CCDSS) in a large university hospital.

Table 1:

AFP – “Test is not compatible with the other markers” Which other markers are referred to here?

The mentioned “biologic invariance rules” are so-called minimal re-testing intervals, therefore the following sentence is not completely correct: “The rules, based on the most recent international guidelines on the prescription appropriateness for laboratory tests” – this covers only the appropriateness concerning the time of testing. Again this statement suggests that all causes for inappropriateness were covered, which is not the case (by far).

The authors focus on reimbursement costs – I am not familiar with the Italian reimbursement system, however, it would be nice to compare these numbers with purely analytical costs (Costs for reagent, instrument and personnel)? Within a hospital setting, there mostly is something like an inner-hospital-reimbursement system, which does not necessarily reflect costs an external, insurancy companies or non-insured patient would have to pay.

Additionally, if possible, try calculating the add-on secondary costs due to inappropriate testing in your patients (costs due to prolonged time to diagnosis and length of stay, unnecessary follow-up diagnostics/therapy; etc) – or at least estimate these numbers within the discussion.

When comparing the amount of inappropriateness to findings of other studies, please keep in mind that the settings of Zhi, Meidani or Feldhammer differed to those in this study. As mentioned above, the wording “inappropriate testing” is misleading, as not all possible causes were included but just a few.

The discussion is way to short. I would urge the authors to discuss also the patient risk of inappropriate lab testing, which is far more important than costs of testing. Additionally, possible causes for inappropriate testing within your health care setting as well as ways of prevention should be discussed.

The conclusion on the other hand is too long – most parts thereof belong to the discussion.

Minor:

Please provide information on where your hospital/lab is located.

The fact that the most inappropriately ordered tests are AFP and Chlamydia PCR is biased by the low number of orders. Please discuss this in the discussion section.

6. PLOS authors have the option to publish the peer review history of their article (what does this mean?). If published, this will include your full peer review and any attached files.

Reviewer #1: No

Reviewer #2: No

---

## [Author Response · Author response to Decision Letter 0]

14 Mar 2020

Dear editor,

I am thankful for your precious suggestions made to improve our work. They were carefully analysed and, as you can see from the attached reviewed document, taken into consideration and completely embraced.

On the basis of your suggestion, we reconsidered the whole paper and the meaning of “inappropriateness” and limited it only to the topic of “test overuse”. In “Discussion” section only comparisons with over-utilization rates of other authors have been included. Even the title has been changed in this light.

Although it is not feasible to apply appropriateness criteria and/or overutilization rules on all possible laboratory tests (over 1000), according to the other authors mentioned, we considered the most frequently performed tests and representative rules (applicable by automatic systems, considering a CCDSS use). We proceeded to specify this aspect in the “Setting” section (row 88).

Despite these limits (better explained in the “Strengths and limitations” section – row 194), the study collected data on a significant number of laboratory tests performed over a long period of time.

The common tests you missed probably fall into the “Complete Blood Count” panel (RBC, Hb, Ht, MCV, MCH, MCHC, RDW, PLTS). However, as you mentioned, we considered it appropriate to better specify the composition of the panel by adding a footnote to the tables and a sentence in the “Materials and methods” section (row 113). 

Finally, we kindly ask you to include in the authors list Professor Maurizio Sanguinetti (President elect and Secretary General of ESCMID, Head of Department of Laboratory Sciences and Infectious Diseases of IRCCS Gemelli Hospital), for his fundamental contribution, as supervisor, in the final stages of this work.

Many thanks for your consideration.

Sincerely,

Dr Andrea Tamburrano

Section of Hygiene - Institute of Public Health

Università Cattolica del Sacro Cuore di Roma

Dear reviewer #1,

I am thankful for your precious revision made to improve our work. Your suggestions were carefully analysed and, as you can see from the attached reviewed document, taken into consideration and completely embraced. 

We report point-by-point answers to your questions:

1) As you suggested, we proceeded to explain the context of the Italian National Health System by specifying in the “Analysis” section that all the treatments were carried out free of charge for the hospitalized patients; hospitalization costs (including laboratory tests) were reimbursed by the Regional Health System according to Medicare Diagnosis Related Groups (MS-DRGs) (row 132).

2) The origin of the requests and the samples is described in the “Setting” section (row 89): only requests from hospital internal departments were monitored.

3) As explained at Point 1, Italian NHS fully covers hospitalization costs according to MS-DRG. This is a flat cost and it’s linked to the main diagnosis and procedures performed during hospitalization. A reduction in inappropriate laboratory tests does not affect the reimbursement costs and generates net savings for the hospital. 

4) Only medical staff can make exam requests in Italian Health System. We proceeded to better specify this information (row 91).

5) No temporary physicians were included in the analysis. Only internal staff could make exam requests through the CPOE login interface.

6) Although your suggestion is acceptable and very interesting, unfortunately, the Local Ethical Committee did not allow us to track information related to physician demography, it’s specific role and the department of origin.

7) As described in “Strengths and limitations” section (row 199) physicians were not alerted and the CCDSS worked silently, without blocking or generating pop-ups. In this way we were able to assess the rate of inappropriateness in laboratory tests overuse without the confounding factor of physician’s awareness of a control system on laboratory requests.

8 and 9) All the examined tests have been ordered and counted as single tests, except for “Complete blood count” in which a panel of tests was ordered and counted in bulk (all together). As you suggested, we better explained this aspect in “Computerized Clinical Decision Support System (CCDSS)” section (row 113). Also, a footnote to the two tables was added.

Following your suggestion, we have reduced the “Conclusion” section by moving part of the text into the “Discussion” section (row 173).

Many thanks for your consideration.

Sincerely,

Dr Andrea Tamburrano

Section of Hygiene - Institute of Public Health

Università Cattolica del Sacro Cuore di Roma

Dear reviewer #2,

I am thankful for your precious revision made to improve our work. Your suggestions were carefully analysed and, as you can see from the attached reviewed document, completely embraced. 

On the basis of your suggestion, we reconsidered the whole paper and the meaning of “inappropriateness” and limited it only to the topic of “test overuse”. Even the title has been changed in this light.

Although it is not feasible to apply appropriateness criteria and/or overutilization rules on all possible laboratory tests (over 1000), according to the other authors mentioned, we considered the most frequently performed tests and representative rules (applicable by automatic systems, considering a CCDSS use). We proceeded to specify this aspect in the “Setting” section (row 88) and to better explain this topic in the “Strengths and limitations” section (row 194).

In this light, as you correctly mentioned, in “Computerized Clinical Decision Support System (CCDSS)” section (row 104) we rephrased the over-utilization criteria specifying the number of biological invariance and incompatibility rules and the preponderance of minimal re-testing intervals criteria. Also, the sentence “The rules, based on the most recent international guidelines on the prescription appropriateness for laboratory tests” was rephrased (row 105).

In Table 1 we provided a better explanation of the rule applied to Alpha Fetoprotein (AFP): “Test cannot be requested with nonspecific tumor markers (CEA, CA125, CA19-9, CA15-3, TPA)”. 

As you suggested, we proceeded to explain the context of the Italian National Health System by specifying in the “Analysis” section that all treatments were carried out free of charge for the hospitalized patients; hospitalization costs (including laboratory tests) were reimbursed by the Regional Health System according to Medicare Diagnosis Related Groups (MS-DRGs) (row 132). Italian NHS fully covers hospitalization costs according to MS-DRG. This is a flat cost and it’s linked to the main diagnosis and procedures performed during hospitalization. A reduction in inappropriate laboratory tests does not affect the reimbursement costs and generates net savings for the hospital.

Although your suggestion to consider analytical and secondary costs is acceptable and very interesting, unfortunately, our data does not allow to make a deep economic evaluation. However, in case of over-utilization criteria and minimal retesting intervals, costs due to a prolonged length of stay, unnecessary diagnostic tests and therapies are substantially negligible.

Following your indications, in the “Discussion” section (row 177) we proceeded to better clarify that comparisons with other authors’ results are affected by a reasonable variability in terms of settings and tests considered. However, only comparisons with over-utilization rates of other authors have been included.

According to your suggestion, we have reduced the “Conclusion” and increased the “Discussion” sections.

As you cited, some tests have a lower number of monthly requests (due to particular clinical needs that motivate the request, related to the epidemiology of the population). Nevertheless, over-utilization rates of each exam were calculated on the total of the requests on a period of 20 months. In this case, rates were sufficiently reliable because they were calculated on 3,520 alpha fetoprotein tests and 1,120 chlamydia trachomatis nucleic acid amplifications (NAATs).

Finally, information on hospital’s location (row 83) was added.

Many thanks for your consideration.

Sincerely,

Dr Andrea Tamburrano

Section of Hygiene - Institute of Public Health

Università Cattolica del Sacro Cuore di Roma

---

## [Decision Letter · Decision Letter 1]

4 May 2020

PONE-D-20-01565R1

Evaluation and cost estimation of laboratory tests overuse through a Computerized Clinical Decision Support System (CCDSS) in a large university hospital.

PLOS ONE

Dear Dr. Tamburrano,

Thank you for submitting your manuscript to PLOS ONE. After careful consideration, we feel that it has merit but does not fully meet PLOS ONE’s publication criteria as it currently stands. Therefore, we invite you to submit a revised version of the manuscript that addresses the points raised during the review process.

The reviewers find that their suggestions are not sufficiently addressed and I concur.

However, as the manuscript has value, I allow another round of revision. I recommend taking the raised issues seriously.

We would appreciate receiving your revised manuscript by Jun 18 2020 11:59PM. To enhance the reproducibility of your results, we recommend that if applicable you deposit your laboratory protocols in protocols.io, where a protocol can be assigned its own identifier (DOI) such that it can be cited independently in the future. For instructions see: http://journals.plos.org/plosone/s/submission-guidelines#loc-laboratory-protocols

We look forward to receiving your revised manuscript.

Kind regards,

Pal Bela Szecsi, M.D. D.M.Sci.

Academic Editor

PLOS ONE

Reviewers' comments:

Reviewer's Responses to Questions

**Comments to the Author**

1. If the authors have adequately addressed your comments raised in a previous round of review and you feel that this manuscript is now acceptable for publication, you may indicate that here to bypass the “Comments to the Author” section, enter your conflict of interest statement in the “Confidential to Editor” section, and submit your "Accept" recommendation.

Reviewer #1: All comments have been addressed

Reviewer #2: (No Response)

Reviewer #3: (No Response)

2. Is the manuscript technically sound, and do the data support the conclusions?

Reviewer #1: Yes

Reviewer #2: Partly

Reviewer #3: Yes

3. Has the statistical analysis been performed appropriately and rigorously? 

Reviewer #1: Yes

Reviewer #2: N/A

Reviewer #3: N/A

4. Have the authors made all data underlying the findings in their manuscript fully available?

Reviewer #1: Yes

Reviewer #2: No

Reviewer #3: Yes

5. Is the manuscript presented in an intelligible fashion and written in standard English?

Reviewer #1: Yes

Reviewer #2: Yes

Reviewer #3: Yes

6. Review Comments to the Author

Reviewer #1: The authors have performed an observational study on the overuse of laboratory tests in a large university hospital.

The manuscript has been revised, and the setting is now clearly described.

Major issues:

1) The Conclusions section is still far too long. Most of it could be transferred to the Discussion section.

Reviewer #2: Several of my suggestion were met inappropriately or not at all:

The mentioned biological invariance rules are basically minimal re-testing interval rules. In the CCDSS-section this should at least be put in brackets behind the term “biological invariance rules” so that the readers are readily able to follow your chain of thoughts.

My suggestion regarding the change of the title was not considered. Once again, I would urge the authors changing it to “selected laboratory tests” or “43 laboratory tests” instead of “laboratory tests”, otherwise readers may assume that all parameters from the labs portfolio were considered in this study, which is not true. Suggestion: “Evaluation and cost estimation of laboratory overuse in 43 parameters by the use of a Computerized Clinical Decision Support System (CCDSS) in a large university hospital.”

The authors misunderstood my suggestion regarding the calculations of costs. Calculating these numbers based on local reimbursement fees may be interesting for the region the study was performed in. However, for all the other readers of PLOS One these numbers are meaningless, since they are not comparable. Therefore, I would suggest a clear and transparent cost calculation based on reagent costs and if possible also personnel and material (blood collection set) costs. As supplemental table a list of single costs for each analyte should be provided. Thereby, readers are able to estimate respective savings in their own setting.

Please also consider the publication rules of PLOS One in this context: “The PLOS Data policy requires authors to make all data underlying the findings described in their manuscript fully available without restriction, with rare exception (please refer to the Data Availability Statement in the manuscript PDF file). The data should be provided as part of the manuscript or its supporting information, or deposited to a public repository. If there are restrictions on publicly sharing data—e.g. participant privacy or use of data from a third party—those must be specified.”

I understand the authors stating that deep economic evaluation of secondary costs is not feasible. In addition thereof however, the statement “costs due to a prolonged length of stay, unnecessary diagnostic tests and therapies are substantially negligible.” is just a subjective belief and imho severely false. – this is just a personal statement and does not have to be discussed in the manuscript.

My suggestion regarding the discussion section was completely ignored: The discussion is way to short. I would urge the authors to discuss also the patient risk of inappropriate lab testing, which is far more important than costs of testing. Additionally, possible causes for inappropriate testing within your health care setting as well as ways of prevention should be discussed.

As for the next revision, please provide a sheet including my suggestion and your direct response to these, so that reviewers do not have to switch between documents and estimate which response belongs to which suggestion.

Reviewer #3: The study by Tamburrano et al addresses a highly relevant question: overuse of laboratory tests. The study focuses on the extra costs generated by overuse of laboratory tests mainly defined as tests repeated within a too short timespan. While this is one important factor causing overuse, another important factor is the increasing number of tests performed per patient usually generated through predefined panels of tests that are easily ordered. Regardless of the cause of overuse, the consequences reach beyond the immediate cost of the tests themselves: Time needed by clinicians to evaluated the results, additional procedures or treatments based on test results outside of reference ranges, sometimes by chance without any connection to the actual morbidity of the patient. A few lines in the discussion section on these points could improve the manuscript.

Specific points:

Line 4 in the “Setting section”: Ematology should be hematology

It is not clear how the mean and SD of the monthly requests and the over-utilization rates in table 2 are calculated. Please specify in the “analysis” section.

7. PLOS authors have the option to publish the peer review history of their article (what does this mean?). If published, this will include your full peer review and any attached files.

Reviewer #1: No

Reviewer #2: No

Reviewer #3: Yes: Henrik L. Jørgensen

---

## [Author Response · Author response to Decision Letter 1]

15 Jun 2020

Dear editor,

I am thankful for your precious suggestions made to improve our work. They were carefully analysed and, as you can see from the attached reviewed document, completely embraced.

Reviewers’ suggestions were now fully addressed, and all the sections of the paper were improved.

Many thanks for your consideration.

Sincerely,

Dr Andrea Tamburrano

Section of Hygiene - Institute of Public Health

Università Cattolica del Sacro Cuore di Roma

 

Dear reviewer #1,

I am thankful for your precious revision made to improve our work. Your suggestions were carefully analysed and, as you can see from the attached reviewed document, completely embraced. 

Following your indications, the “conclusion” section has been shortened and the “discussion section” has been improved.

Many thanks for your consideration.

Sincerely,

Dr Andrea Tamburrano

Section of Hygiene - Institute of Public Health

Università Cattolica del Sacro Cuore di Roma

 

Dear reviewer #2,

I am thankful for your precious revision made to improve our work. Your suggestions were carefully analysed and, as you can see from the attached reviewed document, completely embraced. 

Following your indications, the title was changed, also “discussion” and “conclusions” sections were improved.

We report point-by-point answers to your questions:

• The mentioned biological invariance rules are basically minimal re-testing interval rules. In the CCDSS-section this should at least be put in brackets behind the term “biological invariance rules” so that the readers are readily able to follow your chain of thoughts.

As you requested, we put in brackets the term “minimal re-testing intervals” next to “biological invariance rules” (row 100).

• My suggestion regarding the change of the title was not considered. Once again, I would urge the authors changing it to “selected laboratory tests” or “43 laboratory tests” instead of “laboratory tests”, otherwise readers may assume that all parameters from the labs portfolio were considered in this study, which is not true. Suggestion: “Evaluation and cost estimation of laboratory overuse in 43 parameters by the use of a Computerized Clinical Decision Support System (CCDSS) in a large university hospital.”

As you reported, we changed the title of the paper, following your suggestion, in “Evaluation and cost estimation of laboratory test overuse in 43 commonly ordered parameters through a Computerized Clinical Decision Support System (CCDSS) in a large university hospital”.

• The authors misunderstood my suggestion regarding the calculations of costs. Calculating these numbers based on local reimbursement fees may be interesting for the region the study was performed in. However, for all the other readers of PLOS One these numbers are meaningless, since they are not comparable. Therefore, I would suggest a clear and transparent cost calculation based on reagent costs and if possible also personnel and material (blood collection set) costs. As supplemental table a list of single costs for each analyte should be provided. Thereby, readers are able to estimate respective savings in their own setting.

Please also consider the publication rules of PLOS One in this context: “The PLOS Data policy requires authors to make all data underlying the findings described in their manuscript fully available without restriction, with rare exception (please refer to the Data Availability Statement in the manuscript PDF file). The data should be provided as part of the manuscript or its supporting information, or deposited to a public repository. If there are restrictions on publicly sharing data—e.g. participant privacy or use of data from a third party—those must be specified.”

Regarding the calculation of the costs for each laboratory test, we applied the most reliable and independent calculation available within the Italian healthcare system. In fact, the calculation based on the breakdown of the different items and salaries would return an arbitrary evaluation based on the local hospital contracts with companies. Overall the provided estimates are official evaluation from National Italian authorities and therefore provide the best and standardized estimate we may provide to the PLOS ONE readers.

Different authors analysed this issue and drew similar conclusions.

- Irene et al., for example, say: 

“To attach a cost to a single test or a panel of tests is not a straightforward task due to many factors, including the variability between direct and indirect/overhead costs, proprietary information from vendors on how consumables and equipment costs are broken down, service contracts, and the variability of how each clinical laboratory processes their laboratory tests” 

and 

“Although studies have presented “actual costs” of laboratory tests that are hospital- or province-specific, an “actual cost” of performing a laboratory test varies tremendously across hospitals, cities and provinces. Here, we present the reference median costs (RMC) of commonly ordered laboratory tests in a Canadian setting as a first step towards raising physician awareness in order to help enhance selective and appropriate laboratory test ordering practices.

and 

“Calculating the cost of a laboratory test is complicated as there are a number of fluctuating factors associated with the costs of performing each test sample and the production of the laboratory test result. […] Due to the variability of direct expenses relating to the production of a test result, and variable volumes of tests processed between clinical laboratories, the RMC of each test presented here was determined by compiling price lists of all inclusive indirect costs.”

- Following the same concept, also Sarkar et al. say:

“Approximation of financial cost burden was based on test pricing from two commercial laboratories that might not reflect all commercial laboratories. Hence, the cost analysis should be considered only as an estimation.”

- Finally, Bertrand et al., say:

The measurement of the direct cost of these inappropriate procedures only included the direct cost of the test. For laboratory tests, each item or group of laboratory items was associated with a fixed value (e.g., 10 CHF for the CRP assay). […] The total cost was therefore obtained by adding the set of values associated with the repeated tests. This cost, therefore, did not take into account indirect costs, such as nursing services, time required to complete exams, equipment costs, etc. […] The costs generated by these repeated procedures were estimated on the basis of the unit rates charged at the HUG at CHF 20,000 over one year.”

1. Ma, I., Lau, C. K., Ramdas, Z., Jackson, R., & Naugler, C. (2019). Estimated costs of 51 commonly ordered laboratory tests in Canada. Clinical Biochemistry, 65, 58–60. https://doi.org/10.1016/j.clinbiochem.2018.12.013

2. Sarkar, M. K., Botz, C. M., & Laposata, M. (2017). An assessment of overutilization and underutilization of laboratory tests by expert physicians in the evaluation of patients for bleeding and thrombotic disorders in clinical context and in real time. Diagnosis, 4(1), 21–26. https://doi.org/10.1515/dx-2016-0042

3. Bertrand, J., Fehlmann, C., Grosgurin, O., Sarasin, F., & Kherad, O. (2019). Inappropriateness of Repeated Laboratory and Radiological Tests for Transferred Emergency Department Patients. Journal of Clinical Medicine, 8(9), 1342. https://doi.org/10.3390/jcm8091342

• I understand the authors stating that deep economic evaluation of secondary costs is not feasible. In addition thereof however, the statement “costs due to a prolonged length of stay, unnecessary diagnostic tests and therapies are substantially negligible.” is just a subjective belief and imho severely false. – this is just a personal statement and does not have to be discussed in the manuscript.

Following your indications, the statement was reported in the correct formula in the “conclusions” section (row 222-224)

• My suggestion regarding the discussion section was completely ignored: The discussion is way to short. I would urge the authors to discuss also the patient risk of inappropriate lab testing, which is far more important than costs of testing. Additionally, possible causes for inappropriate testing within your health care setting as well as ways of prevention should be discussed.

Following your indications, the issues you raised were included and argued in the “discussion” section (row 195-203).

Many thanks for your consideration.

Sincerely,

Dr Andrea Tamburrano

Section of Hygiene - Institute of Public Health

Università Cattolica del Sacro Cuore di Roma

Dear reviewer #3,

I am thankful for your precious revision made to improve our work. Your suggestions were carefully analysed and, as you can see from the attached reviewed document, completely embraced. 

Following your indications, the issues you raised were included and argued in the “discussion” section (row 195-203).

As you reported, “ematology” was corrected in “hematology”. 

Finally, a description of the descriptive statistics performed is reported in the “analysis” section (row 127-129).

Many thanks for your consideration.

Sincerely,

Dr Andrea Tamburrano

Section of Hygiene - Institute of Public Health

Università Cattolica del Sacro Cuore di Roma

---

## [Decision Letter · Decision Letter 2]

3 Jul 2020

PONE-D-20-01565R2

Evaluation and cost estimation of laboratory test overuse in 43 commonly ordered parameters through a Computerized Clinical Decision Support System (CCDSS) in a large university hospital.

PLOS ONE

Dear Dr. Tamburrano,

Thank you for submitting your manuscript to PLOS ONE. After careful consideration, we feel that it has merit but does not fully meet PLOS ONE’s publication criteria as it currently stands. Therefore, we invite you to submit a revised version of the manuscript that addresses the points raised during the review process.

Please correct the few remaining issues mentioned by the reviewers.

It will not require renewed review.

We look forward to receiving your revised manuscript.

Kind regards,

Pal Bela Szecsi, M.D. D.M.Sci.

Academic Editor

PLOS ONE

Reviewers' comments:

Reviewer's Responses to Questions

**Comments to the Author**

1. If the authors have adequately addressed your comments raised in a previous round of review and you feel that this manuscript is now acceptable for publication, you may indicate that here to bypass the “Comments to the Author” section, enter your conflict of interest statement in the “Confidential to Editor” section, and submit your "Accept" recommendation.

Reviewer #1: (No Response)

Reviewer #2: (No Response)

Reviewer #3: All comments have been addressed

2. Is the manuscript technically sound, and do the data support the conclusions?

Reviewer #1: Yes

Reviewer #2: Partly

Reviewer #3: Yes

3. Has the statistical analysis been performed appropriately and rigorously? 

Reviewer #1: Yes

Reviewer #2: N/A

Reviewer #3: Yes

4. Have the authors made all data underlying the findings in their manuscript fully available?

Reviewer #1: Yes

Reviewer #2: Yes

Reviewer #3: Yes

5. Is the manuscript presented in an intelligible fashion and written in standard English?

Reviewer #1: Yes

Reviewer #2: Yes

Reviewer #3: Yes

6. Review Comments to the Author

Reviewer #1: The Authors have performed a study on the over-utilization of laboratory tests. This is an important issue due to the ever-increasing demand for laboratory services and diminishing health care resources.

Minor issues:

1) The provider of the Prometeo Appropriatessa Software should be provided.

Reviewer #2: The autors have met most of my comments. However, I still believe that the Discussion is too thin and that the authors could dive deeper into patient safety risks of inappropriate lab testing - actually this should be the main motivation for improvement actions.

Reviewer #3: (No Response)

7. PLOS authors have the option to publish the peer review history of their article (what does this mean?). If published, this will include your full peer review and any attached files.

Reviewer #1: No

Reviewer #2: No

Reviewer #3: No

---

## [Author Response · Author response to Decision Letter 2]

14 Jul 2020

Dear editor,

I am thankful for your precious suggestions made to improve our work. They were carefully analysed and, as you can see from the attached reviewed document, completely embraced.

Reviewers’ suggestions were now fully addressed, and all the sections of the paper were improved.

The provider of the Prometeo Appropriatezza Software has been provided in parentheses (row 97).

The discussion section has been improved by adding a discussion on patient safety risks related to inappropriate routine tests (row 203-210 and 214-215).

Many thanks for your consideration.

Sincerely,

Dr Andrea Tamburrano

Section of Hygiene - Institute of Public Health

Università Cattolica del Sacro Cuore di Roma

---

## [Editor Report · Decision Letter 3]

22 Jul 2020

Evaluation and cost estimation of laboratory test overuse in 43 commonly ordered parameters through a Computerized Clinical Decision Support System (CCDSS) in a large university hospital.

PONE-D-20-01565R3

Dear Dr. Tamburrano,

We’re pleased to inform you that your manuscript has been judged scientifically suitable for publication and will be formally accepted for publication once it meets all outstanding technical requirements.

Kind regards,

Pal Bela Szecsi, M.D. D.M.Sci.

Academic Editor

PLOS ONE
---

## [Editor Report · Acceptance letter]

27 Jul 2020

PONE-D-20-01565R3 

Evaluation and cost estimation of laboratory test overuse in 43 commonly ordered parameters through a Computerized Clinical Decision Support System (CCDSS) in a large university hospital. 

Dear Dr. Tamburrano:

I'm pleased to inform you that your manuscript has been deemed suitable for publication in PLOS ONE. Congratulations! Your manuscript is now with our production department. 

Kind regards, 

on behalf of

Dr. Pal Bela Szecsi 

Academic Editor

PLOS ONE